# miRNA Molecules—Late Breaking Treatment for Inflammatory Bowel Diseases?

**DOI:** 10.3390/ijms24032233

**Published:** 2023-01-23

**Authors:** Ioanna Aggeletopoulou, Athanasia Mouzaki, Konstantinos Thomopoulos, Christos Triantos

**Affiliations:** 1Division of Hematology, Department of Internal Medicine, Medical School, University of Patras, 26504 Patras, Greece; 2Division of Gastroenterology, Department of Internal Medicine, Medical School, University of Patras, 26504 Patras, Greece

**Keywords:** microRNAs, inflammatory bowel disease, ulcerative colitis, Crohn’s disease, therapeutic targets

## Abstract

MicroRNAs (miRNAs) are a group of non-coding RNAs that play a critical role in regulating epigenetic mechanisms in inflammation-related diseases. Inflammatory bowel diseases (IBDs), which primarily include ulcerative colitis (UC) and Crohn’s disease (CD), are characterized by chronic recurrent inflammation of intestinal tissues. Due to the multifactorial etiology of these diseases, the development of innovative treatment strategies that can effectively maintain remission and alleviate disease symptoms is a major challenge. In recent years, evidence for the regulatory role of miRNAs in the pathogenetic mechanisms of various diseases, including IBD, has been accumulating. In light of these findings, miRNAs represent potential innovative candidates for therapeutic application in IBD. In this review, we discuss recent findings on the role of miRNAs in regulating inflammatory responses, maintaining intestinal barrier integrity, and developing fibrosis in clinical and experimental IBD. The focus is on the existing literature, indicating potential therapeutic application of miRNAs in both preclinical experimental IBD models and translational data in the context of clinical IBD. To date, a large and diverse data set, which is growing rapidly, supports the potential use of miRNA-based therapies in clinical practice, although many questions remain unanswered.

## 1. Introduction

### 1.1. Inflammatory Bowel Diseases Overview

Inflammatory bowel diseases (IBDs) are chronic inflammatory diseases of the gastrointestinal tract that primarily include ulcerative colitis (UC) and Crohn’s disease (CD) [1]. UC is associated with diffuse mucosal inflammation and ulceration extending for a variable distance from the rectum to the caecum. CD is primarily characterized by transmural inflammation occurring at any site in the gastrointestinal tract; the terminal ileum and colon are most commonly affected. IBDs have become a public health challenge worldwide [2], as their incidence and prevalence have increased significantly over the past decade in both Western and Eastern countries [3,4], necessitating the development of new treatment strategies. Besides the direct effects of IBDs on the gastrointestinal tract, extra-intestinal manifestations are also common. These complications are mainly due to the chronic and systemic inflammatory state that IBDs induce by disrupting various signaling pathways, which in turn alter the expression of regulatory mediators, such as cytokines and microRNAs (miRNAs) [5,6].

### 1.2. IBDs Pathogenesis

The pathogenesis of IBDs remains elusive, but significant progress has been made in recent years in understanding the pathophysiology of these diseases. The mechanisms associated with the pathophysiology and development of IBDs mainly include dysregulated immune responses, environmental changes, gut dysbiosis, and disease-related genetic alterations [7,8,9,10,11,12,13]. In addition to the above factors, miRNAs—a class of small, single-stranded, non-coding ribonucleic acid (RNA) molecules of approximately 22 nucleotides—have also been associated with the pathophysiology of IBDs [14].

### 1.3. miRNA Function

miRNAs have been shown to be involved in the regulation of gene expression by targeting mRNA, causing post-transcriptional gene silencing or mRNA degradation, and thus controlling protein production [15]. miRNAs are highly involved in various biological functions, such as cell proliferation, maturation and differentiation, signal transduction, cell apoptosis, modulation of chronic inflammation and carcinogenesis, and they control various cellular and metabolic pathways [16]. Several applications for predicting mRNA/miRNA interactions have been developed, enabling better selection and interpretation of miRNA target prediction analysis tools and contributing to the understanding of how miRNAs achieve their regulatory effects [17]. Since the disrupted regulation and function of miRNAs is strongly associated with intestinal diseases, the development of miRNA-based therapeutic strategies is being investigated. Recent studies have shown that miRNAs are differentially expressed in autoimmune diseases; in particular, miRNAs have been associated with critical inflammatory pathways involved in the pathogenesis of IBD [14,18]. In the present review, we summarized recent advances in the exploration of the regulatory role of miRNAs in the development of inflammation, intestinal barrier integrity, and fibrosis in IBD and provided an outlook on the challenges of introducing innovative therapeutic strategies based on miRNA-related signal transduction, both in experimental and clinical IBD, paving the way for the development of new drugs.

## 2. miRNA Overview

The first miRNA was discovered in Caenorhabditis elegans by Lee et al. [19] in 1993 and designated as lin-4, which was able to regulate the translation of lin-14 via an antisense interaction RNA-RNA. These molecules are expressed in most eukaryotes, including humans [20]. miRNAs are single-stranded RNAs of 19 to 24 nucleotides and were originally considered to be developmental regulatory genes encoding small antisense RNA products [19]. According to miRBase v.22, the human genome encodes approximately 2600 mature miRNAs, and according to GENCODE data v.29, more than 200,000 transcripts [21]. Most miRNAs are transcribed from DNA into primary miRNAs (pri-miRNAs), which then transform into precursor miRNAs (pre-miRNAs) and mature miRNAs [22]. Suppression of protein production occurs via two mechanisms (Figure 1).

First, the mature miRNA is formed by two-step cleavage of the pri-miRNA, which interacts with the RNA-induced silencing complex (RISC). The miRNA binds to the target mRNA and negatively regulates its expression. The complementarity between the miRNA and the mRNA target determines the proper silencing mechanism: degradation of the mRNA target or translational silencing. The end result of these processes is a decrease in the amount of protein produced [22]. For this reason, impaired expression of miRNAs can lead to aberrant cellular functions and alter downstream gene regulation and related signaling pathways. Most miRNAs inhibit the expression of target mRNAs through their interaction with the 3′-untranslated region (3′-UTR) [23], but miRNAs can also interact with other regions, including the 5′-UTR, coding sequences, and gene promoters [24]. Multiple mRNAs can be affected by a single miRNA, and a given mRNA is often regulated by multiple miRNAs. Conversely, multiple miRNAs may also be required for translational blockade of a given mRNA. As a result, miRNAs may be responsible for disruptions in gene expression, particularly in chronic inflammatory diseases, such as IBDs. At the same time, miRNAs have been shown to conditionally upregulate gene expression [25].

There is increasing evidence for the role of miRNAs as pro- and anti-inflammatory molecules, oncogenes, or tumor inhibitors. Therefore, targeting miRNA-related functional biomolecules in specific cell types and systems and in various experimental models may elucidate their precise function in pathogenic processes.

Overall, recent studies have highlighted the critical role of miRNAs in the pathogenesis of IBD and associated chronic inflammatory complications, highlighting these molecules as potential candidates for disease prognosis and indicators of treatment suitability, and suggesting their inclusion in the therapeutic armamentarium of IBD.

## 3. Potential Therapeutic Application of miRNAs

Biological agents have been shown to be effective therapies against inflammatory mediators involved in the pathophysiology of IBD. These therapies include anti-tumor necrosis factor (TNF) agents, monoclonal antibodies, and targeted therapies [26,27,28]. Although biologic therapy has important advantages in terms of specificity compared with glucocorticoids or other immunosuppressive drugs, it also has disadvantages, such as the absence of responses or the occurrence of various side effects. miRNA-based treatment strategies offer a new perspective for the future treatment of patients with IBD [29]. Recent findings have demonstrated the important regulatory role of miRNAs in IBD. The potential of developing miRNAs targeting IBD-related genes is appealing. However, to meet this challenge, miRNA molecules must be developed with high specificity, efficiency, and safety when delivered to inflamed tissues. The development of off-target side effects remains a major concern, as altering one miRNA activity can affect multiple downstream target genes and signaling pathways. The delivery system for miRNAs also remains a major challenge. The expression and activity of miRNAs may be restricted to a specific cell context; due to the different profiles of miRNAs, a miRNA may be expressed differently in different cell types and exert different functions. Therefore, changes in the expression of such miRNAs may be protective in certain cells, whereas they may play detrimental roles in other cell types. Accurate delivery of the specific miRNA to the target cells may help to eliminate the side effects in vivo, which is also a major challenge in the application of miRNA treatment.

Another obstacle that miRNA therapeutics must overcome is that of oligonucleotide drugs, such as digestion of RNAs in the blood, excretion by the kidneys, blockage by the vascular barrier, and low uptake by certain cells [30,31]. Nanoparticles manufactured for transport of miRNAs into target cells could be an effective option for this process. In addition, miRNAs consisting of the same sequence have different names due to their genomic location [32]. Further research on the genomic location of miRNAs may provide insights into the impaired regulation of molecular mechanisms and the development of IBD. Another challenge is to deliver the RNA complexes through the vascular endothelial barrier into the target tissue. Another hurdle is the mechanism by which miRNAs control protein levels through complementary base pairing with the coding regions of target genes.

miRNA-based therapeutics include two different strategies: miRNA mimics and miRNA antagonists. miRNA antagonists include antisense oligonucleotides that preferentially suppress the “seed regions” of miRNA to activate blockade of downstream signaling pathways [33]. The use of miRNA antagonists aims to restore impaired function of an mRNA target resulting from overexpression of a miRNA [34]. However, treatment with antagonists is associated with difficulties in terms of specificity, hepatotoxicity, and the development of side effects [33,35].

miRNA mimics, or agomirs, can restore the decreased miRNA expression caused by upregulated targeting [34]. However, treatment with miRNA mimics is also associated with difficulties related to the required dosage and incorporation of miRNA into the fully functional RISC complex [35].

## 4. Therapeutic Use of miRNAs in the Context of Inflammatory Responses

miRNAs are important molecules associated with the development of IBD and act as inflammatory inhibitors or activators by positively or negatively regulating immune signaling pathways associated with IBD. At the same time, miRNAs are considered critical modulators of innate and adaptive immunity, including cell differentiation and cell signaling. Innate immunity and, in particular, processes associated with signaling cascades, including those of toll-like receptors (TLRs) and nucleotide-binding oligomerization domain-containing (NOD)-like receptors (NLRs), are strongly modulated by miRNAs [36]. The signaling pathways most involved in these processes are those of the nuclear factor kappa-light-chain-enhancer of activated B cells (NF-κBs) and mitogen-activated protein (MAP) kinase [36]. In parallel, miRNAs play an important role in the maturation and stimulation of B and T cells; the differentiation of T helper 1 (Th1), Th2, and Th17 cells and the homeostasis of T regulatory cells (Tregs) are highly modulated by these molecules [18,37].

The main function and targets of each miRNA have been investigated by in vitro studies; these studies using cell lines and/or experimental animal models evaluate the role of each miRNA by stimulating or suppressing its expression using agomiRs and antagomiRs, which are commercially available or synthesized. In addition, experimental animal models have been developed to study the effects of adding or silencing a single miRNA gene in vivo. These experimental models that mimic human IBD include the dextran sulphate sodium (DSS) model and the trinitrobenzene sulfonic acid (TNBS) model, which induce colitis in rats and mice and are widely accepted models to study intestinal inflammation in experimental IBDs [38].

The most studied miRNAs and their potential role as therapeutic targets for the treatment of IBD are reviewed below.

### 4.1. miR-21

miR-21 is a widely known pro-oncogenic gene that is highly expressed in various cancers, including breast, colon, pancreatic, and gastric cancers, and in recent years, has also been implicated in the pathogenesis of autoimmune diseases [39]. miR-21 is expressed at low levels in resting T cells and antigen-presenting cells (APCs); however, cell stimulation results in significantly higher expression of miR-21 [40].

#### 4.1.1. UC

Increased expression of miR-21 was detected in active UC colon tissue compared with healthy controls [41,42]. These data were confirmed by a subsequent study that revealed higher miR-21 levels in lamina propria macrophages and T cells from UC patients [43]. miR-21 knockout (KO) mice were less susceptible to DSS-induced colitis than their wild-type (WT) counterparts, and antibiotic treatment resulted in loss of protection. Simultaneous housing of healthy and miR-21 KO mice attenuated this effect and made the WT mice less susceptible. Treatment with DSS resulted in more severe colitis symptoms in mice colonized with WT fecal homogenate than in mice colonized with miR-21 KO fecal homogenate, suggesting that miR-21 is a key player in the susceptibility to intestinal inflammation through changes in the gut microbiota [44]. In a study by Lu et al. [45], significantly increased miR-21-5p expression was detected in the sera of UC patients and in the colon tissue of rats with DSS-induced colitis. Transfection of a miR-21-5p inhibitor into LPS-induced RAW264.7 cells resulted in lower interleukin (IL)-6 and TNF-α levels and inhibition of pro-apoptotic markers, indicating the potential role of miR-21-5p blockade as an anti-inflammatory target in human UC [45].

#### 4.1.2. CD

Increased expression of miR-21 was detected in UC compared with CD patients, suggesting that miR-21 is more specific for UC immunopathogenesis than a prognostic marker for inflammation [43]. Another study showed that miR-21 significantly decreased in peripheral blood mononuclear cells (PBMCs) from CD patients compared to healthy controls [46].

In conclusion, the data demonstrated that suppression of miR-21 is a potential therapeutic target in UC patients because it inhibits important pro-inflammatory cytokines and critically modulates the homeostasis of the gut microbiota.

### 4.2. miR-124

#### 4.2.1. UC

miR-124 is an evolutionarily highly conserved miRNA and one of the most abundantly expressed miRNAs in the central nervous system; it has anti-inflammatory activity and exerts numerous biological functions, including autophagy, cell proliferation, regulation of immunity, and neuronal differentiation [47].

Signal transducer and activator of transcription 3 (STAT3), an important signaling pathway that mediates immune suppression in the tumor microenvironment, is targeted and suppressed by miR-124, which contributes to the regulation of T-cell functions [47]. A decrease in miR-124, followed by an upregulation of STAT3, has been demonstrated in colon samples from pediatric UC patients [48]. The protective effect of nicotine against DSS-induced colitis was induced by expression of miR-124 and downstream inhibition of STAT3, indicating the potential role of miR-124/STAT3 as a key player in the therapeutic armamentarium of UC [49].

#### 4.2.2. CD

The role of miR-124 as a pro-inflammatory molecule has been proposed in CD patients; in this case, miR-124 targets the aryl hydrocarbon receptor (AhR), which is downregulated in the gut of patients with IBD. An AhR ligand called TCDD stimulates miR-124, leading to amelioration of DSS-induced experimental colitis by regulating Th17 and Treg differentiation or inducing secretion of the anti-inflammatory cytokine IL-22 [50,51]. An inverse relationship between miR-124 and AhR levels was found in intestinal epithelial cells and colon tissues from patients with active CD. However, data have shown that the use of anti-miR-124 treatment alleviated intestinal inflammation by inhibiting AhR in experimental TNBS-induced colitis [52]. The same study also showed higher miR-124 levels, followed by decreased AhR levels, in Caco-2 and HT-29 cell lines after stimulation of the inflammatory response by LPS in vitro [52]. Another study demonstrated more severe TNBS colitis in an AhR KO mouse model compared to WT mice and higher miR-124a levels after TNBS exposure compared to WT mice, confirming the above findings [53].

### 4.3. miR-146

#### 4.3.1. UC

The miR-146 family consists of two genes, miR-146a and miR-146b, known for their anti-inflammatory effects [54]; in particular, they act as negative feedback regulators of innate immunity through their contribution to the TLR/NF-κB signaling pathway [55].

Data on the role of miR-146 in intestinal inflammation are conflicting. In the context of IBD, miR-146a has been found to be upregulated in inflamed intestinal tissue from CD and UC patients compared with healthy or non-inflamed mucosa [56,57]; on the other hand, its expression is reduced in Tregs from UC patients [58]. Given the suppressive effect of miR-146a, alteration of its expression could potentially lead to impairment of pathogenic Th1 responses and autoimmunity in IBDs.

In parallel, miR-146a^-/-^ mice showed resistance to the DSS-induced colitis model by blocking genes related to the intestinal barrier [59]. However, overexpression of miR-146b protected against DSS-induced colitis by stimulating NF-κB signaling and improving epithelial barrier function [60]. Based on these findings, miR-146a administration via extracellular vesicles was investigated in rats with TNBS-induced colitis. The results showed higher expression of miR-146a in the colon, which led to alleviation of colitis by attenuating inflammation mediated via MAPK and NF-κB signaling pathways [61]. Oral administration of miR-146b-bearing nanoparticles protected miR-146b-deficient mice from DSS-induced colitis [62]. Inhibition of colitis was defined by decreased expression of the pro-inflammatory cytokines IL-1β and TNF-α in M1 macrophages. In contrast, the number of M2 macrophages increased after miR-146b nanoparticle administration, highlighting the important contribution of miR-146b in controlling the transition of macrophages from a pro-inflammatory M1 to an anti-inflammatory M2 phenotype [62]. Conflicting data emerged from studies reporting that suppression of miR-146a by a synthetic inhibitor [63] or by oral administration of the antidiabetic drug vildagliptin [64] resulted in amelioration of experimentally induced colitis in rats.

#### 4.3.2. CD

The critical role of miR-146b has also been demonstrated in the pathogenesis of CD, as it has been associated with intestinal barrier impairment. Li et al. [65] demonstrated that a long noncoding RNA (lncRNA), MALAT1, is abnormally downregulated in the intestinal mucosal tissues of CD patients and in DSS-induced colitis mice; the mechanism of action of MALAT1 is to sequester miR-146b-5p and maintain the expression of apical junction complex [AJC] proteins. In parallel, MALAT1 KO mice were hypersensitive to DSS-induced acute colitis. Suppression of miR-146b-5p ameliorated experimental colitis, suggesting a pro-inflammatory function [65].

In conclusion, there are controversial data on the precise role of the miR-146 family in the pathogenesis of IBD. Most importantly, the improvement of experimental colitis in mice after miR-146b nanoparticle administration is probably related to the overall restoration of disturbed homeostasis rather than a direct effect. Moreover, an opposing role of the two mature sequences of miR-146 has been postulated. Further research is needed to develop targets against the miR-146 family, focusing on their function as anti-inflammatory molecules or negative regulators of mucosal barrier function in the gut.

### 4.4. miR-155

#### 4.4.1. UC

miR-155 is a multifunctional miRNA that targets more than 25 genes and is instrumental in modulating inflammatory diseases [66]. It is responsible for regulating homeostasis and inhibiting oncogenesis and is highly expressed in the thymus and spleen [67]. Recent studies have shown that miR-155 is one of the most important potential therapeutic targets for the treatment of IBD. Takagi et al. [41] first pointed out the importance of miR-155 (together with miR-21) in the pathophysiology of UC; in particular, upregulation of miR-155 was demonstrated in the sigmoid colon of UC patients. These results were confirmed by a subsequent study that found decreased expression of forkhead box O3 (FOXO3a) in the colon tissue of active UC patients and in HT29 cells treated with TNF-α, demonstrating that miR-155 significantly decreases FOXO3a expression in human colon epithelial cells [68]. Another study showed that FOXO3a deficiency resulted in severe gut inflammation in vivo, demonstrating a TNF-α-dependent role of miR-155 in the gut [69].

Blockade of the miR-155/NF-κB axis is a promising treatment strategy for IBD. The NF-κB pathway is an important mediator of inflammatory responses, and overexpression of miR-155 has been shown to trigger NF-κB activation in mouse macrophages [70]. miR-155-mediated reduction of FOXO3a has been shown to positively regulate nucleotide-binding domain-like receptor protein 3 (NLRP3) inflammasome [71]. This finding is supported by data showing that induced NF-κB expression leads to higher NLRP3 expression [72]. Blockade of miR-155 in lipopolysaccharide (LPS)-activated RAW 264.7 cells resulted in downregulation of inflammatory cytokines by reducing pNF-κB and NLRP3-related proteins [73]. Based on these findings, the use of the antimalarial drug artesunate was evaluated, and the results showed that it suppressed NF-κB signaling by suppressing miR-155 in a TNBS-induced colitis model and in LPS-induced RAW 264.7 mouse macrophages, promoting its potential use as a therapeutic strategy in UC patients [74]. Similarly, the use of chlorogenic acid was tested in LPS-induced RAW 264.7 cells and DSS-induced colitis, and the results showed that it exhibited protective effects against colitis by blocking miR-155-dependent activation of the NF-κB/NLRP3 pathway and downregulating the expression of the pro-inflammatory cytokines IL-1β and IL-18 [73]. Anti-miR-155-5p treatment has been shown to inhibit granulocyte colony-stimulating factor (G-CSF), a regulator of granulopoiesis secreted by macrophages during the acute inflammatory response [75]. The alkaloid sinomenine has been shown to decrease the expression of miR-155 and several related inflammatory cytokines in TNBS-induced colitis in mice, acting as an anti-inflammatory factor [76].

In addition to the role of miR-155 in innate immunity, it also contributes critically to the control of adaptive immunity by modulating the differentiation of CD4+ T cells and Tregs [77]. Singh et al. [78] reported that miR-155 KO mice had fewer Th17 cells in mesenteric lymph nodes in response to DSS-induced colitis and lower levels of pro-inflammatory cytokines. The therapeutic effect of miR-155 antagomirs to maintain the balance between Th17 and Treg cells was tested in C57BL6/J mice [79]. The results showed that miR-155 antagomir improved the disease activity index and led to a decrease in Th17 cells and a concomitant downregulation of the cytokines IL-17A and IL-6, but an increase in Tregs, IL-10, and TGF-B1 in the mesenteric lymph nodes, suggesting that maintaining the balance between Th17 and Treg cells is a means to attenuate colitis [79].

JARID2, another downstream target of miR-155, has been shown to suppress the expression of Est-1, a negative modulator of Th17 cells in DSS-induced colitis, and promote the expression of IL-6, IL-17, and IL-23, as well as the maturation of Th17 cells [80,81]. In addition, miR-155 is essential for the generation and activity of follicular T helper cells [82]. Knockdown of miR-155 resulted in the attenuation of colitis symptoms and the reduction of clinical score and disease severity by reducing Th1, Th17, CD11b+, and CD11c+ cells [78]. Another important factor in the development of inflammation in IBD is the miR-155/Src homology 2 domain-containing inositol 5’-phosphatase-1 (SHIP-1) signaling pathway. miR-155 appears to contribute to the pathogenesis of colitis via suppression of SHIP-1 expression, while restoration of SHIP-1 has been shown to alleviate intestinal inflammation [83].

Another interesting finding was the reversal of genomic instability and inflammation and the acceleration of colon healing in cultured intestinal epithelial cells and in mice with DSS-induced colitis by targeted inhibition of miR-23a and miR-155 [84]. miR-155 antagomir also reduced intestinal barrier impairment and resolved inflammation in a DSS-induced colitis model by inducing the expression of hypoxia-inducible factor 1 (HIF-1), a barrier protective factor involved in mucosal inflammation [85].

Finally, Pathak et al. [86] reported that inhibition of miR-155 in intestinal myofibroblasts from UC patients resulted in decreased cytokine production and higher expression of suppressor of cytokine signaling 1 (SOCS1), whereas silencing of SOCS1 in intestinal control myofibroblasts greatly increased the production of pro-inflammatory cytokines. Thus, suppression of miR-155 may represent a novel therapeutic target for preventing impaired cytokine secretion by intestinal myofibroblasts, thereby contributing to the alleviation of inflammation.

#### 4.4.2. CD

In addition to the effects of miR-155 in UC and experimental colitis, data have also shown its effects in CD. A meta-analysis demonstrated a number of miRNAs that are differentially expressed between UC and CD; among these, miR-155-5p was upregulated in UC compared with CD [87]. Guz et al. [88] reported that miR-155-5p was significantly overexpressed in inflamed CD ileum and colon tissues compared with healthy tissues.

miR-155 has been shown to increase the expression of the pro-inflammatory cytokine TNF-α and decrease the expression of the anti-inflammatory cytokine IL-10 in CD24hiCD27+ B cells, allowing its use in the development of IL-10-producing B cell-based strategies to improve CD outcome [89].

In conclusion, blockade of miR-155 leads to attenuation of intestinal inflammation in vitro and in experimental models, suggesting its use as an anti-inflammatory agent for the treatment of IBD. The important role of miR-155 in regulating the TNF-α cascade makes this molecule a potential target for the treatment of IBD.

### 4.5. miR-144/451

Although the biological activity of miR-144/451 has been studied mainly in erythropoiesis and tumorigenesis, recent studies have been conducted on immune responses. A recent study showed that miR144/451 expression was reduced in dendritic cells (DCs) from both patients with IBD and experimental DSS-colitis mice compared with controls [90]. Human DCs showed decreased expression of miR144/451 after LPS activation [90]. miR-144/451 KO resulted in severe colitis in a DSS-induced colitis model, whereas DCs derived from the periphery and mesenteric lymph nodes secreted higher levels of pro-inflammatory cytokines and co-stimulatory molecules compared with WT mice [90]. Moreover, miR-144/451 KO DCs transplantation exacerbated DSS-induced colitis [90]. In the experimental model with bone marrow transplantation, miR-144/451 KO transplantation of bone marrow exacerbated DSS-induced colitis [90]. Treatment of mice with miR-144/451 nanoparticles showed a protective effect on DSS-induced colitis [90]. These results suggest that control of miR-144/451 expression in patients with IBD could be considered as a therapeutic tool for IBD treatment by targeting the miR144/451-interferon regulatory factor (IRF5) pathway [90].

## 5. Therapeutic Use of miRNAs in the Context of Intestinal Epithelial Barrier Function

The intestinal epithelial barrier is a physical and biochemical barrier that regulates interactions between components of the lumen, such as the intestinal microbiota and mucosal immune system, and maintains intestinal homeostasis [91]. Disruption of this barrier is a critical mechanism mediating the pathophysiology of IBD [1]. Therefore, there is increasing evidence for the role of miRNA-mediated control of intestinal permeability.

miR-93 has been shown to have a protective role for the intestinal barrier by targeting protein tyrosine kinase 6 (PTK6); PTK6 induces nuclear accumulation of FoxO1, which in turn downregulates expression of the tight junction (TJ) protein claudin-3 [92]. Haines et al. [92] showed that PTK6-null mice exhibited improvement in intestinal barrier function; however, the use of miR-93 mimics attenuated TNF-α/IFN-γ-mediated disruption of the intestinal epithelial barrier through PTK6 downregulation in vitro.

miR-122a was strongly associated with TNF-α-dependent TJ permeability, as suppression of miR-122a was shown to inhibit TJ permeability in vitro [93]. In parallel, TNF-α-dependent overexpression of miR-122a increased intestinal permeability in vivo [93]. Overexpression of miR-122 was able to reduce NOD2 in intestinal epithelial tissue, resulting in the blockade of apoptosis and intestinal barrier damage [94]. miR-200c-3p negatively regulates the expression of occludin in UC patients and in DSS-induced experimental colitis; consequently, antagomiR-200c, an antagonist of miR-200c-3p, inhibited occludin reduction in vitro and in the intestinal tissues of experimental colitis models, preserving the integrity of the TJ barrier [95].

The use of miR-155 is associated with intestinal barrier dysfunction and has been shown to reduce TJ protein expression in DSS-induced colitis [85]. Hypoxia-inducible factor 1a (HIF-1a), which is involved in signaling pathways regulating anti-inflammatory responses, was downregulated by miR-155, whereas treatment with miR-155 antagomir increased HIF-1α levels [85]. These data highlight the role of miR-155 in DSS-induced colitis by inducing intestinal barrier dysfunction and suppressing the HIF-1α/TFF-3 axis [85]. In parallel, miR-155 has been implicated in the modulation of TJs, such as occludin, claudin-1, and myosin light chain kinase (MLCK) [96]. Increased miR-155 levels have been shown to control elevated levels of IL-13 by downregulating IL-13R1, the major receptor subunit for IL-13, in UC [97]. Elevated IL-13 levels in Th2-mediated UC have been associated with epithelial barrier dysfunction through alterations in claudin-2 expression and a higher risk of apoptosis [98,99]. Expression of E-cadherin, a key molecule of adherens junctions, is also downregulated by miR-155, further reducing mucosal stability in UC patients and increasing the risk of cancer metastasis [88].

Activation-induced cytidine deaminase (AID), a key molecule required to promote immunoglobulin (Ig) class switch recombination, is targeted by miR-155; blockade of miR-155 results in AID and gut immune barrier enhancement [100,101].

Results on the protective role of miRNAs on gut barrier function showed that miR-200b alleviates gut inflammation and permeability by targeting myosin light chain kinase; this interaction leads to a reduction in TNF-α-associated TJ dysfunction [102]. A significant inverse relationship was found between hsa-miR-200c-3p and immune-related IL-8 and between hsa-miR-200c-3p and CDH11, a key player in gut barrier function [103]. These results suggest that hsa-miR-200c-3p directly controls the expression of IL-8 and CDH11 in inflamed UC mucosa by counteracting inflammatory IL-8 activity or downregulating the NF-κB response to TLR4 activation [103].

Another characteristic of patients with IBD is the presence of high levels of reactive oxygen species (ROS) due to the constant stimulation of macrophages or the ability of the organism to sense and respond to changes in oxygen in intestinal tissue. Several miRNAs have been associated with the modulation of nitric oxide synthase-2 (NOS2) in IBD. Stimulation of the NO pathway by miR-21, miR-126, miR-146a, miR-221, and miR-223 led to senescence in neighboring epithelial cells by increasing heterochromatin protein 1 γ (HP1γ) [104]. Regarding the sensing of ambient oxygen in intestinal tissues, HIF is an important player in regulating barrier integrity; thus, increasing miR-320a led to improved barrier function in T84 cells [105]. The development of new strategies to measure oxygen levels in the form of free radicals and a gaseous state in inflamed intestinal tissue could be a valuable tool to monitor disease progression.

In conclusion, there is growing evidence that miRNAs play a critical role in regulating the permeability of the intestinal epithelial barrier. Therefore, new treatment strategies should focus on targeting miRNAs to maintain and protect the proper functioning of the intestinal epithelial barrier.

## 6. Therapeutic Use of miRNAs in the Context of Intestinal Fibrosis

Intestinal fibrosis is a common complication of IBDs due to the persistent immune-mediated intestinal inflammation that occurs in these diseases [106]. Specifically, impaired regulation of intestinal tissue repair leads to excessive deposition of extracellular matrix (ECM) in the intestinal layers, resulting in the development of tissue fibrosis [107].

This process is likely triggered by the activation of inflammatory signaling pathways following tissue injury; however, there are emerging data suggesting that it is a self-perpetuating, inflammation-independent process [108]. Until recently, the role of miRNAs in the pathogenesis of intestinal fibrosis was unknown. Despite increasing research on the role of miRNAs in IBD-associated intestinal inflammation, which may lead to the development of miRNA-based treatments, their impact on intestinal fibrosis remains to be elucidated. The role of miRNAs in fibrogenesis is unclear, as they act as both profibrotic and antifibrotic factors [109].

In addition to the role of miR- 155-5p in intestinal inflammation, this molecule has been found to contribute strongly to the fibrogenesis of IBD, together with miR-146a-3p. Notably, an inverse association between miR-155 and E-cadherin was detected in CD patients, but not in UC, suggesting that this complex is involved in the epithelial–mesenchymal transition (EMT), which plays a crucial role in the development of intestinal fibrosis [110]. Moreover, miR-155 was been found in greater amounts in intestinal tissues from CD patients with fibrotic strictures than in patients without fibrosis [111]. Data in vivo and in vitro have shown that miR-155 is responsible for the repression of high mobility group box transcription factor 1 (HBP1), a critical suppressor of the Wnt/β-catenin signaling cascade, leading to pathway stimulation and a consequent increase in fibrosis-associated markers. In parallel, the use of a miR-155 mimic in experimental TNBS-induced colitis resulted in significant intestinal fibrosis, whereas the use of a miR-155 suppressor alleviated fibrosis [111]. These results indicate that miR-155 is a potential candidate for the treatment of both intestinal fibrosis and inflammation in patients with IBD.

However, further research on the inhibition of profibrogenic miRNAs is essential, as the development of anti-miRNA therapies targeting intestinal fibrosis is far from being achieved.

On the other hand, several miRNAs, including miR-29a, miR-29b, and miR-29c, have shown an antifibrotic phenotype. Members of the miR-29 family were found to be suppressed in non-inflamed mucosal tissue in strictured compared to non-strictured regions of CD patients. miR-29b was found to be a negative modulator of both type I α2 and 3 α1 collagen in CD fibroblasts, and overexpression of miR-29b reversed TGF-β1-mediated collagen expression in CD-related fibrosis [112]. The antifibrotic role of miR-29b was also demonstrated by its indirect induction of the antifibrotic protein myeloid cell leukemia (MCL)-1 [113]. Specifically, miR-29b negatively regulates collagen synthesis and positively regulates MCL-1, which in turn inhibits intestinal fibrosis [113].

## 7. Other

It has been proposed to use miR-195 as a biomarker for UC because lower levels of miR-195 lead to higher expression of Smad7 and upregulation of p65 and activator protein 1 (AP-1) [114]. Gene correlation network analysis revealed a close relationship between miR-200c and IBD, suggesting that miR-200c reduces cellular inflammation and NLRP3 inflammasome-related cell pyroptosis in vitro and ameliorates DSS-induced IBD symptoms in vivo by targeting NIMA-related kinase 7 (NEK7) [115]. These results may partially explain the mechanism of steroid resistance in UC patients, which is a major obstacle to effective UC treatment [114].

## 8. miRNA Gene-Associated Regulatory Networks

Hundreds of miRNAs have now been described, and each is capable of influencing multiple gene transcripts. miRNAs are members of complex gene regulatory networks (GRNs) consisting of feedback and feedforward loops [116,117]. Specific sub-circuits are evolutionarily conserved and are referred to as network motifs [117]. Modulation of transcription in parallel with miRNA-mediated gene regulation represents a recurrent network motif and reinforces gene regulation in mammalian genomes [116]. With the development of computational and, more recently, high-throughput experimental methods to identify miRNA targets, miRNA circuits have been described to be associated with the development of inflammatory states: NF-κB and hepatocyte nuclear factor-4α (HNF-4α) circuits [118,119]. In the NF-κB circuit, stimulation of tyrosine protein kinase Src triggers an NF-κB-mediated inflammatory response that leads to downregulation of miRNA let-7a and induction of IL-6 [118]. This process generates a stable positive feedback loop across multiple cell divisions [118]. Similarly, the HNF-4α circuit includes miR-124, STAT3, IL-6R, miR-629, and miR-24 and is critical for proper hepatocyte and liver function [119]. Figure 2 shows selected miRNA-mediated regulatory circuits.

## 9. miRNA-Associated Drugs Being Tested in Clinical Trials

### 9.1. ABX464 for UC

Although miR-124 has been considered a pro-inflammatory molecule that induces intestinal inflammation in CD patients and in CD experimental models, its anti-inflammatory role in UC has paved the way for the development of a miRNA-based drug. ABX464 (Abivax, Paris, France) is an orally administered drug originally produced as a suppressor of HIV replication that promotes expression of miR-124 from the miR-124.1 genomic locus [120,121]. miR-124 is formed by splicing of the long noncoding RNA lncRNA 0599-205 [121]. ABX464 acts through its binding to the cap-binding complex, an important element of RNA biogenesis, by triggering splicing of lncRNA 0599-205 and inducing miR-124 expression [120]. The use of ABX464 was tested in a phase IIa proof-of-concept study evaluating its safety and efficacy in patients with moderate-to-severe UC [122]. UC patients who completed the induction phase were eligible for a long-term extension phase. The registration numbers for these studies were NCT03093259 [123] for the induction phase and NCT03368118 [124] for the long-term phase. ABX464 has been shown to interact with the cap-binding complex and lead to upregulation of miR-124, a unique RNA splicing molecule with anti-inflammatory activity. This process results in DSS-induced modulation of proinflammatory cytokine production. Preclinical studies showed that ABX464 attenuated DSS-induced colitis in mice, provided long-term protection, and reduced levels of miR-124 after treatment discontinuation [125].

The primary endpoint in the induction phase was safety, which was assessed by the frequency of adverse events. Efficacy endpoints were achievement of clinical remission at week 8 compared with placebo, Mayo clinic score (MCS) and partial MCS (pMCS) assessment from baseline (CFB) to week 8, endoscopic remission and improvement, histopathologic assessment by Geboes score from CFB to week 8, changes in fecal calprotectin, and expression of miR-124 [122]. In the long-term extension phase, the primary endpoint was the long-term safety of ABX464 use [122]. Results showed that clinical remission and clinical response were achieved in 35% and 70%, respectively, of UC patients in the ABX464 group at week 8, compared with 11% and 33%, respectively, in the placebo group at week 8 [122]. Improvement in endoscopic markers and treatment response was seen in 50% and 10%, respectively, of patients in the ABX464 group versus 11% and 11%, respectively, in the placebo group. Patients also showed endoscopic and histologic improvements. Efficacy results did not reach statistical significance, but there was a positive trend [122].

A phase 2b clinical trial (NCT04023396) evaluating the long-term efficacy and safety of ABX464 as maintenance therapy in patients with moderate-to-severe UC is expected to confirm these results [126].

### 9.2. ABX464 for CD

Abivax has also initiated a phase IIa study (NCT03905109) evaluating the safety and efficacy of ABX464 in patients with moderate-to-severe active CD, who have had inadequate response or intolerance to prior treatment with amino-salicylates, immunosuppressants, biologics, and/or corticosteroids [127]. This study has not yet been completed.

## 10. MicroRNA-Based Delivery Systems

MicroRNA delivery systems have been studied for a variety of pathologies. Viral and nonviral miRNA delivery systems have been described (Figure 3).

### 10.1. Viral miRNA-Based Delivery Systems

Viral vectors are capable of precisely delivering genes into target cells. Several viral vectors have been developed to mediate RNA interference because they can transfer genes to different tissues and organs, resulting in long-term expression of the target gene. The particular properties of each viral vector make them suitable for specific delivery conditions. Adeno-associated viral vectors, lentiviral vectors, retroviral vectors, and bacteriophages are the most commonly used viral miRNA-mediated delivery systems [128].

### 10.2. Non-Viral miRNA-Based Delivery Systems

In addition to viral ones, non-viral delivery systems have also been described. In multicellular organisms, maintenance of cellular homeostasis requires proper functioning of long-distance intercellular communication.

#### 10.2.1. Cell-Derived Membrane Vesicles

There is growing evidence that cell-to-cell communication can occur through extracellular vesicles (EVs) [129]. EVs play an important role in intercellular communication and have been used as biomarkers and drug carriers [130]. Three main types of EVs have been described based on specific features, such as molecular profile and intracellular origin: microvesicles, exosomes, and apoptotic bodies. Microvesicles are released from the cell membrane of different cell types under certain pathological and physiological conditions [131]. Exosomes are membrane-bound EVs produced in the endosomal unit of most eukaryotic cells. These membrane vesicles are involved in intercellular communication, antigen presentation, and mRNA and miRNA shuttling and are derived from late endosomes [132]. There is increasing evidence for the role of exosomes in mediating intercellular communication through the carriage of miRNAs and subsequent protection of small RNAs from RNases [133].

Data have shown that miRNA-carrying natural EVs can improve IBD in experimental models. Specifically, EVs derived from the hookworm Nippostrongylus provided mice with protection against TNBS-induced intestinal inflammation [134]. In parallel, administration of milk-derived EVs containing miR-148 ameliorated DSS-induced colitis in mice by interfering with the TLR4–NF-κB pathway [135].

Administration of human umbilical cord mesenchymal stem cell (hucMSC)-derived exosomes was protective against DSS-induced colitis because miR-378a-5p delivered by EVs suppressed NLRP3 inflammasome assembly and the resulting cleavage of caspase-1 in vitro, resulting in decreased IL-1β and IL-18 maturation [136]. Similarly, HucMSC-derived exosomes containing miR-326 ameliorated DSS-induced inflammation by blocking NF-κB signaling [137]. Another study that also investigated the effect of hucMSC-Ex on alleviating IBD in mice showed that hucMSC-Ex improved intestinal lymphatic drainage and suppressed lymphangiogenesis and macrophage infiltration [138]. Mechanistically, miR-302d-3p was found at high levels in hucMSC-Ex and contributed significantly to the inhibition of lymphangiogenesis by targeting Fms-related receptor tyrosine kinase 4 (FLT4) [138]. In parallel, AKT phosphorylation was blocked and vascular endothelial growth factor receptor 3 (VEGFR3) was decreased, suggesting that the regulatory role of hucMSC-Ex in lymphangiogenesis via the miR-302d-3p/VEGFR3/AKT axis mediates amelioration of IBD [138]. Another study addressed the role of hucMSC-Ex in IBD treatment via the caspase (casp) 11/4 pathway [139]. The results showed that hucMSC-Ex treatment ameliorated DSS-induced colitis by blocking casp11/4-induced macrophage pyroptosis, while hucMSC-Ex carrying miR-203a-3p.2 suppressed casp4-induced THP-1 macrophage pyroptosis, suggesting a potential use of this molecule in IBD treatment [139].

The role of bone mesenchymal stem cell (BMSC)-derived exosomes carrying miR-539-5p in alleviating IBD was investigated, and the results showed a suppressive effect on pyroptosis through the NLRP3/caspase-1 pathway and subsequently on IBD progression [140].

Apoptotic bodies are “little sealed sacs” that contain substances and information from apoptotic cells. Apoptotic bodies are larger in diameter and are responsible for recruiting phagocytes to neighboring apoptotic cells, leading to their clearance [141]. Platelets derived from bone marrow megakaryocytes contribute to the maintenance of vascular integrity and hemostasis [142].

#### 10.2.2. Lipid-Based Nanoparticles

Lipid nanocarriers are flexible and versatile nanoparticles that can be effectively chemically modified to conjugate with targeting molecules and fluorescent probes to serve as safe nucleic acid carriers in vivo [143]. Numerous studies have investigated the application of cationic liposomes as miRNA transporters in vivo. To date, a large number of cationic lipids have been prepared for nucleic acid drug delivery; however, this method has the disadvantage of low delivery efficiency, which limits their clinical use. Therefore, novel lipids have been prepared and new methods for preparing lipid nanocomplexes have been developed. Polymer-based therapeutics include polymeric nanoparticles, polymeric micelles, dendrimers, polyplexes, polymersomes, and polymer–lipid hybrid systems.

#### 10.2.3. Polymeric Vectors/Dendrimer-Based Vectors

Polymeric nanoparticles are commonly used as drug carriers and can be functionalized with ligands to improve targeting to cell surface receptors [144]. Several polymer-based nanoparticles have been approved for clinical use [145]. Polyethylenimines carry a large number of amine groups and are positively charged. Therefore, they are able to bind to small RNAs and form nanoscale groups that provide protection against RNA degradation and enhance intracellular release and uptake into cells [146]. However, the use of polyethylenimine has the disadvantage of toxicity, which limits its use in current clinical practice. Dendrimers are three-dimensional, monodisperse, spherical nanopolymeric materials with a branched tree-like structure. Specific features of various dendrimers give them unique advantages, such as self-assembly, polyvalence, electrostatic interactions, low cytotoxicity, chemical stability, and solubility [147].

#### 10.2.4. Inorganic Material-Based Delivery Systems

Inorganic compound-based materials, such as gold nanoparticles, mesoporous silicon, silver nanoparticles, graphene oxide, and Fe_3_O_4_-mediated nanoparticles, have been developed as miRNA delivery vectors and are widely used in nanotechnology [144]. Functional complexes can be easily attached to the surface of gold nanoparticles and used as miRNA carriers [148].

#### 10.2.5. Three-Dimensional Scaffold-Based Delivery Systems

Finally, three-dimensional (3D) scaffold-based delivery systems that circumvent mechanical barriers and allow spatiotemporal control can precisely control the therapeutic effect of miRNA [149]. To date, several 3D scaffolds, including hydrogels, electrospun fibers, and other highly porous or spongy 3D scaffolds, have been introduced for miRNA delivery [149].

## 11. Conclusions and Prospects

In recent years, the role of miRNAs as diagnostic tools or therapeutic targets has been widely investigated. With respect to UC and CD, the study of miRNAs has provided valuable insights into understanding disease pathogenesis and developing alternative therapeutic strategies. The ability of miRNAs to post-transcriptionally modulate the expression of numerous genes has highlighted them as attractive molecules for drug development. Although much progress has been made during these years, several obstacles to effective miRNA treatment remain. In particular, the molecular networks underlying miRNA-mediated posttranscriptional modulation are still largely unclear. Dysregulations of the immune system or impaired gut barrier function that characterize UC and CD can be modulated at the miRNA level. Recent findings have shed light on how miRNAs can be transported through EVs to restore barrier disruption, alleviate IBD symptoms, and improve disease progression. The recent Food and Drug Administration (FDA) approval of four siRNA-based therapeutics paved the way for the use of RNA molecules for the therapy of chronic diseases [150]. Despite the existence of numerous miRNAs that contribute to the pathogenesis of IBD, the precise role of most miRNAs in IBD is still unclear; therefore, further research is needed to determine the holistic regulatory role of miRNAs as a therapeutic modality in IBD. Well-designed therapeutic trials investigating the complex miRNA networks and their target genes are necessary to develop novel interventions aimed at alleviating disease symptoms and maintaining clinical remission.

## Figures and Tables

**Figure 1 ijms-24-02233-f001:**
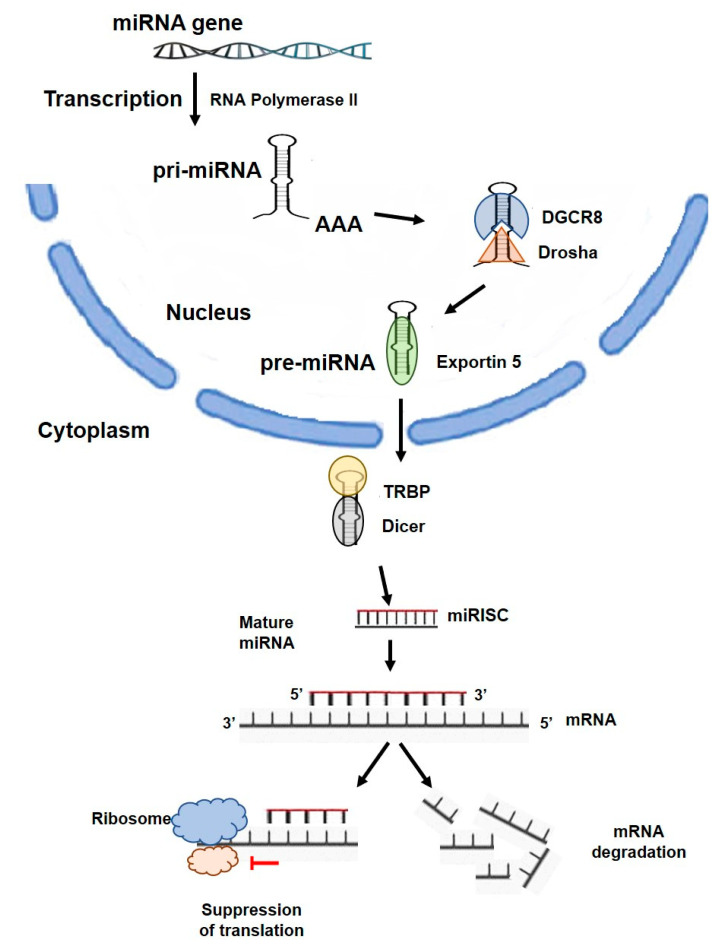
miRNA structure and biogenesis. miRNA genes are mainly transcribed by RNA polymerase II in the nucleus to produce pri-miRNA transcripts. The microprocessor complex, composed of the RNase III enzyme Drosha and the dimeric RNA-binding protein DGCR8, produces the pre-miRNA precursor product by cleavage of pri-miRNA. The pre-miRNA is transferred to the cytoplasm by exportin 5 (XPO5). A complex composed of the Rnase III enzyme Dicer and the transactivation response element RNA-binding protein (TRBP) cleaves the pre-miRNA into a mature double-stranded miRNA. A mature miRNA is then incorporated into the miRNA-associated multiprotein RNA-induced silencing complex (mi-RISC). The mature miRNA then binds to complementary regions in the target mRNA and acts as a guide through base pairing with the mRNA to modulate its expression. In most cases, the mature miRNA binds to the 3′-untranslated sequences (3′-UTR) of specific mRNAs via partially complementary sequences and inhibits the translation of the mRNAs into protein. If there is high complementarity between the miRNA and the mRNA, this leads to cleavage of the target mRNA.

**Figure 2 ijms-24-02233-f002:**
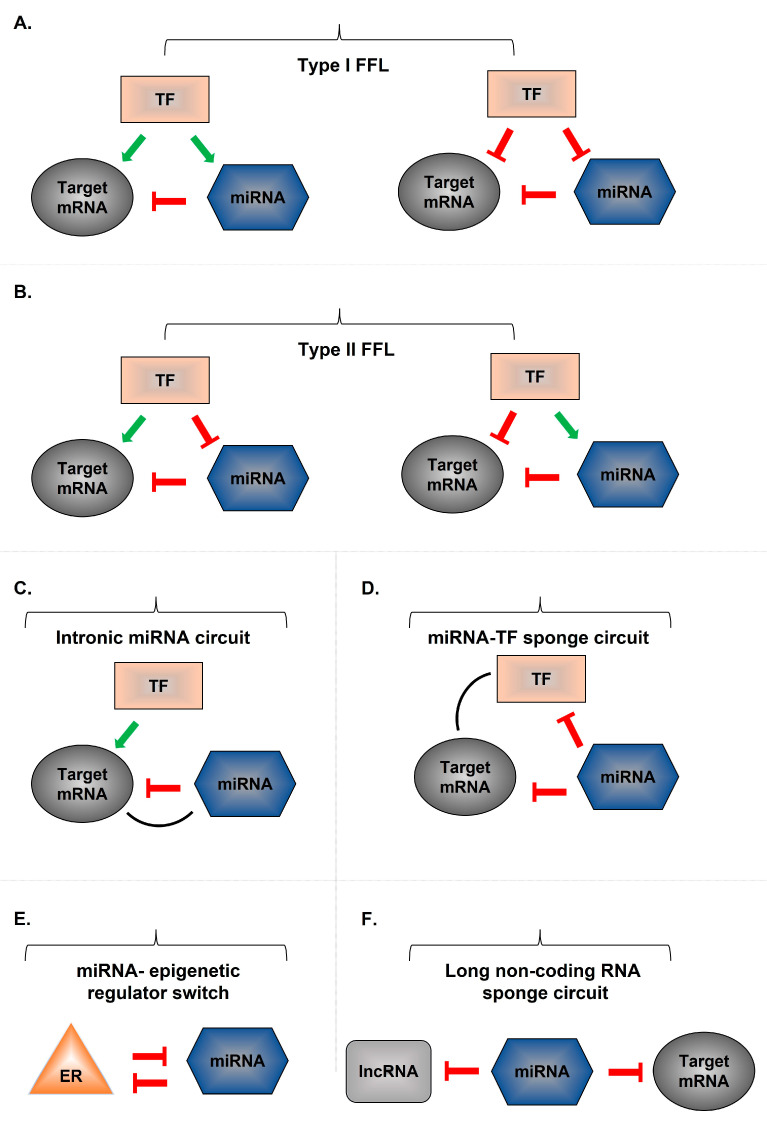
**Multiple miRNA-mediated circuits.** A common miRNA-mediated feed-forward loop (FFL) includes a master transcription factor (TF) regulating a miRNA and an mRNA target gene. miRNA-mediated FFLs are divided into type I (incoherent) or type II (coherent) FFLs, which are determined by the association of the miRNA transcription and the target gene (co-regulation or opposing regulation by the same TF). (**A**) In Type I circuits (incoherent action), TFs positively regulate miRNA and their target mRNAs and aim to define and maintain protein homeostasis, especially in cell populations that have high sensitivity to the target mRNA. (**B**) In Type II circuits (coherent action), transcriptional activation or repression (positive or negative FFL) of a target mRNA by a TF is allowed, resulting in synergistic miRNA expression. In the case of mRNA repression, the TF downregulates the target mRNA and upregulates the miRNA. In the case of mRNA upregulation, the TF upregulates the mRNA, resulting in synergistic miRNA suppression. (**C**) Intronic miRNA-mediated self-loop (iMSL) consists of a TF that regulates both the miRNA and a host mRNA gene encoded by a single genomic site. In this loop, the miRNA is usually placed in an intron of the host gene, and transcription occurs at the same time; thus, the TF regulates the miRNA and the host gene in the same way. (**D**) In miRNA-modulated FFL, the miRNA acts as a master regulator and controls a TF. In this circuit, the interaction between the TF and the target can be both activating and repressive. (**E**) In the double negative feedback loop in the presence of an epigenetic regulator, the miRNA targets the epigenetic regulator, while the expression of the same miRNA is controlled by the epigenetic regulator. In this circuit, both the miRNA regulation of a target gene and the regulation of an epigenetic regulator of the miRNA are negative. (**F**) In the sponge circuit, the long non-coding RNAs (lncRNAs) control target genes through a miRNA-dependent mechanism. The miRNA controls both the target mRNA and the lncRNAs through the miRNA recognition elements (MREs).

**Figure 3 ijms-24-02233-f003:**
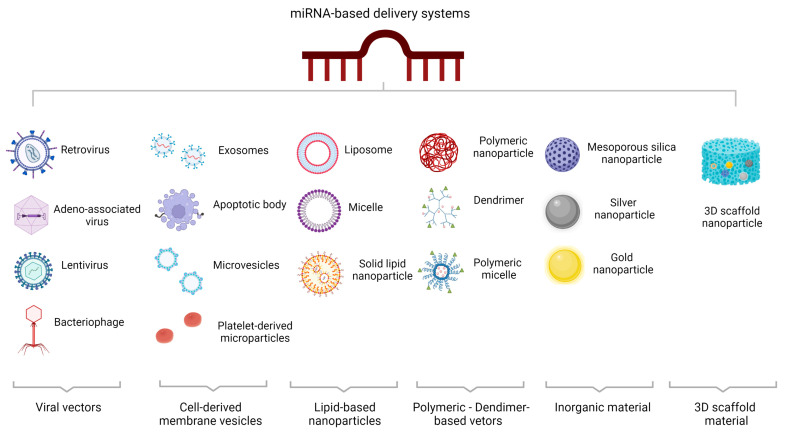
Viral and non-viral miRNA-based delivery systems. Nonviral miRNA-based delivery systems are divided into several categories: cell-derived membrane vesicles, lipid-based nanoparticles, polymeric vectors/dendrimer-based vectors, inorganic material-based delivery systems, and 3D scaffold-based delivery systems. This figure was generated using BioRender, Available online: https://biorender.com (accessed on 29 September 2022).

## Data Availability

Not applicable.

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
