# Peer review of "miRNA Molecules—Late Breaking Treatment for Inflammatory Bowel Diseases?"

_ijms, 2023, doi:10.3390/ijms24032233_

Round 1

Reviewer 1 Report

Dear authors,

Please consider the following comments to improve the content of your manuscript before publication. 

The work is highly relevant from a therapeutic point of view, since it puts into context what is known up to now about miRNAs in inflammatory bowel diseases.

The writing seems adequate to me with relevant information, as well as easy reading. The images are didactic, especially Figure 2.

In my opinion, the article has the necessary arguments for its publication as it is, they only need to make the following corrections:

Line 2: putting the “m” in “miRNA” in lowercase is the correct terminology.

Lines 29, 48, 51, 67, 76, 140, 166, 173, 180: correct the term MiRNAs by miRNAs (correct).

Lines 205, 206, 208, 215, 239, 241, 270, 314, 316, 331, 368, 373, 392, 408, 414, 418, 498, 519: correct the term MiR by miR (correct).

From Line 43 onwards the text is not justified, correct it.

Line 64: write in italics the scientific name Caenorhabditis elegans.

Line 78: write Drosha or DROSHA instead of drosha.

Line 81: write Dicer or DICER instead of dicer.

Lines 153, 198, 327, 416, 484, 512, 689, 690: write in italics in vivo.

Lines 193, 265, 396, 397, 413, 415, 421, 484, 511, 680: write in italics in vitro.

Line 184: remove a space between the words “particular, process”

Line 336: remove a space between the words “reducing pNF-B”

Line 409: remove a space between the words “(PTK6); PTK”

Line 410: leave the abbreviation TJ (tight junction) defined from this line since it is the first time it appears, instead of line 424.

Line 482: remove a space between the words “crucial role”

Lines 552-565: the paragraph is in italics, correct to normal font.

Line 595 (Figure 2): the titles of each outline (A-F) should be where the brace opens (at the top) of each outline.

Lines 647-649: the figure caption should be below Figure 3, not on top.

Line 674: write in italics the scientific name Nippostrongylus.

REFERENCES 

In general, most of the references should be corrected, the name of the journals is not abbreviated, the name of the journal and/or volume is missing, as well as italicizing all the scientific names, the font is not the same as that of the entire text. Please review and adapt as requested by the journal.

Please amend the requested comments and submit the revision file.

Author Response

Reviewer 1

Dear authors,

Please consider the following comments to improve the content of your manuscript before publication. 

The work is highly relevant from a therapeutic point of view, since it puts into context what is known up to now about miRNAs in inflammatory bowel diseases.

The writing seems adequate to me with relevant information, as well as easy reading. The images are didactic, especially Figure 2.

In my opinion, the article has the necessary arguments for its publication as it is, they only need to make the following corrections:

Comment 1: Line 2: putting the “m” in “miRNA” in lowercase is the correct terminology.

Response to comment 1: We have revised as suggested.

Comment 2: Lines 29, 48, 51, 67, 76, 140, 166, 173, 180: correct the term MiRNAs by miRNAs (correct).

Response to comment 2: We have revised as suggested.

Comment 3: Lines 205, 206, 208, 215, 239, 241, 270, 314, 316, 331, 368, 373, 392, 408, 414, 418, 498, 519: correct the term MiR by miR (correct).

Response to comment 3: We have revised as suggested.

Comment 4: From Line 43 onwards the text is not justified, correct it.

Response to comment 4: We have revised it as indicated. (Lines 51-54)

Comment 5: Line 64: write in italics the scientific name Caenorhabditis elegans.

Response to comment 5: We have revised as suggested.

Comment 6: Line 78: write Drosha or DROSHA instead of drosha.

Response to comment 6: We have revised it accordingly.  

Comment 7: Line 81: write Dicer or DICER instead of dicer.

Response to comment 7: We have revised it accordingly.  

Comment 8: Lines 153, 198, 327, 416, 484, 512, 689, 690: write in italics in vivo.

Response to comment 8: We have revised as suggested.

Comment 9: Lines 193, 265, 396, 397, 413, 415, 421, 484, 511, 680: write in italics in vitro.

Response to comment 9: We have revised as suggested.

Comment 10: Line 184: remove a space between the words “particular, process”

Response to comment 10: We have revised it accordingly.  

Comment 11: Line 336: remove a space between the words “reducing pNF-B”

Response to comment 11: We have revised it accordingly.  

Comment 12: Line 409: remove a space between the words “(PTK6); PTK”

Response to comment 12: We have revised it accordingly.  

Comment 13: Line 410: leave the abbreviation TJ (tight junction) defined from this line since it is the first time it appears, instead of line 424.

Response to comment 13: We have revised it as suggested.

Comment 14: Line 482: remove a space between the words “crucial role”

Response to comment 14: We have revised it accordingly.  

Comment 15: Lines 552-565: the paragraph is in italics, correct to normal font.

Response to comment 15: Nippostrongylus

Comment 16: Line 595 (Figure 2): the titles of each outline (A-F) should be where the brace opens (at the top) of each outline.

Response to comment 16: We have revised as suggested.

Comment 17: Lines 647-649: the figure caption should be below Figure 3, not on top.

Response to comment 17: We have revised all figure captions as suggested.

Comment 18: Line 674: write in italics the scientific name Nippostrongylus.

Response to comment 18: We have revised it accordingly.  

Comment 19: In general, most of the references should be corrected, the name of the journals is not abbreviated, the name of the journal and/or volume is missing, as well as italicizing all the scientific names, the font is not the same as that of the entire text. Please review and adapt as requested by the journal.

Response to comment 19: We have revised the References section as suggested in order to adhere Journal’s requirements.   

Reviewer 2 Report

This MS reviews aspects of the potential role of miRNA in IBD

SPECIFIC COMMENTS

1. In the INTRO, UC can extend to the caecum (pancolonic) but can also be left sided or distal. Suggest adding "extending for a variable distance from the rectum to the caecum"

2. The INTRO is one single long paragraph. Suggest to format into smaller paragraphs to aid readability

3. Bacterial (and other names of organisms) should be in standard format (italics)

4. The section from line 116 and after covers general aspects of IBD, rather than linking in with the preceding text. Suggest to move accordingly

5. Line 127 and follows sets the scene for the review. this is appropriate to be at the end of the INTRO

Some of the text in this area of the MS does not flow well (parts don't fit together well) and could be improved with careful revision ensuring enhanced focus and clarity

6. further any reference to the current work should be in the past tense, not the present tense

7. Please ensure that person first concepts are followed throughout.  FOR EXAMPLE: "IBD patients" should be "patients with IBD"

8. Some referencing is absent or misplaced. For instance, the following statement lacks a reference (which must be placed at the end of the sentence): Increased expression of miR-21 was detected in active UC colon tissue compared with 213 healthy controls. 

9. When using an author name and et al, the relevant reference must be directly after et al (and not some time later). "..... Lu et al (43)...." as one example

10. section 8 is all in italics unnecessarily

11. Terms should be provided in full first: Nuclear Factor (NF)...

12. The legend for FIG 3 gives no explanation of the details of the figure

13. Some other parts also need improved formatting. Page 17 is essentially one single long paragraph

14. Please review all REFERENCES to ensure that they follow the journal requirements fully. Several (e.g. References 15, 18 & 21) have full journal titles rather than standard abbreviations. Reference 27 has no journal title.

15. 

Author Response

Reviewer 2

This MS reviews aspects of the potential role of miRNA in IBD.

Comment 1: In the INTRO, UC can extend to the caecum (pancolonic) but can also be left sided or distal. Suggest adding "extending for a variable distance from the rectum to the caecum".

Response to comment 1: We have revised as suggested.

Comment 2: The INTRO is one single long paragraph. Suggest to format into smaller paragraphs to aid readability.

Response to comment 2:  We have divided the Introduction section into subsections as suggested.

Comment 3: Bacterial (and other names of organisms) should be in standard format (italics).

Response to comment 3: We have revised as suggested throughout the manuscript.

Comment 4: The section from line 116 and after covers general aspects of IBD, rather than linking in with the preceding text. Suggest to move accordingly.

Response to comment 4: This section has been moved to the Introduction section as suggested.  

Comment 5:  Line 127 and follows sets the scene for the review. this is appropriate to be at the end of the INTRO. Some of the text in this area of the MS does not flow well (parts don't fit together well) and could be improved with careful revision ensuring enhanced focus and clarity.

Response to comment 5: The aim of the study has been moved to the end of the Introduction section, as suggested. The Introduction section has been re organized to achieve better focus and clarity as suggested.

Comment 7: further any reference to the current work should be in the past tense, not the present tense.

Response to comment 7: We have revised as suggested.

Comment 8: Please ensure that person first concepts are followed throughout.  FOR EXAMPLE: "IBD patients" should be "patients with IBD"

Response to comment 8: We have revised as suggested throughout the manuscript.

Comment 9: Some referencing is absent or misplaced. For instance, the following statement lacks a reference (which must be placed at the end of the sentence): Increased expression of miR-21 was detected in active UC colon tissue compared with 213 healthy controls. 

Response to comment 9: We have added the quoted references as proposed.

Comment 10: When using an author name and et al, the relevant reference must be directly after et al (and not some time later). "..... Lu et al (43)...." as one example.

Response to comment 10: We have revised as suggested throughout the manuscript.

Comment 11: section 8 is all in italics unnecessarily.

Response to comment 11: We have revised as suggested.

Comment 12: Terms should be provided in full first: Nuclear Factor (NF)...

Response to comment 12: There is no abbreviation “NF” referring to nuclear factor in the current manuscript. The only reference of NF is into the acronyms TNF, referring to tumor necrosis factor, HNF-4α referring to hepatocyte nuclear factor-4α and NF-κB referring to  nuclear factor kappa-light-chain-enhancer of activated B cells, which are already displayed.

Comment 13: The legend for FIG 3 gives no explanation of the details of the figure.

Response to comment 13: We have elaborated on the legend of figure 3. However, there is detailed description of the figure 3 contents on the manuscript. 

Comment 14: Some other parts also need improved formatting. Page 17 is essentially one single long paragraph.

Response to comment 14: The section 10 titled “MicroRNA-based delivery systems” has been re organized and divided into subsections, as suggested.

Comment 15: Please review all REFERENCES to ensure that they follow the journal requirements fully. Several (e.g. References 15, 18 & 21) have full journal titles rather than standard abbreviations. Reference 27 has no journal title.

Response to comment 15: We have revised the References section as suggested in order to adhere Journal’s requirements.   

Round 2

Reviewer 2 Report

Thank you for improvements

section 7 is still a single long paragraph, that could also be improved with reformatting

Author Response

Reviewer 2

Thank you for improvements.

Comment 1: section 7 is still a single long paragraph, that could also be improved with reformatting.

Response to comment 1:  We thank the reviewer for this comment. The section 7 has been re organized in order to be more comprehensible, as suggested. In the revised manuscript we have moved the part referring to exosomes use in IBD from the section “7. Other” to the section “10.2.1. Cell-derived membrane vesicles”. Furthermore, we have moved the part referring to the miR-144/451 from the section “7. Other” to the section “4.    Therapeutic use of miRNAs in the context of inflammatory responses” and now has been designated as “4.5. miR-144/451”.